# DSB: Dynamic Sliding Block Scheduling for Diffusion LLMs

**Lizhuo Luo** [* 1]  **Shenggui Li** [* 1]  **Yonggang Wen** [1]  **Tianwei Zhang** [1]

## Abstract

Diffusion large language models (dLLMs) have emerged as a promising alternative for text generation, distinguished by their native support for parallel decoding. In practice, block inference is crucial for avoiding order misalignment in global bidirectional decoding and improving output quality. However, the widely-used fixed, predefined block (naive) schedule is agnostic to semantic difficulty, making it a suboptimal strategy for both quality and efficiency: it can force premature commitments to uncertain positions while delaying easy positions near block boundaries. In this work, we analyze the limitations of naive block scheduling and disclose the importance of dynamically adapting the schedule to semantic difficulty for reliable and efficient inference. Motivated by this, we propose **Dynamic Sliding Block (DSB)**, a training-free block scheduling method that uses a sliding block with a dynamic size to overcome the rigidity of the naive block. To further improve efficiency, we introduce **DSB Cache**, a training-free KV-cache mechanism tailored to DSB. Extensive experiments across multiple models and benchmarks demonstrate that DSB, together with DSB Cache, consistently improves both generation quality and inference efficiency for dLLMs. Code is released at https://github.com/lizhuo-luo/DSB.

## 1. Introduction

Diffusion models have emerged as a leading paradigm for image (Podell et al., 2023; Xie et al., 2024) and video (Blattmann et al., 2023; Kong et al., 2024) generation, and recent progress suggests that masked diffusion models can be similarly competitive for text generation. Different from autoregressive (AR) language models (OpenAI et al., 2024; Qwen et al., 2025) that generate tokens strictly left-to-right,

---
[*]Equal contribution  [1]Nanyang Technological University. Correspondence to: Tianwei Zhang <tianwei.zhang@ntu.edu.sg>.

*Proceedings of the $43^{rd}$ International Conference on Machine Learning*, Seoul, South Korea. PMLR 306, 2026. Copyright 2026 by the author(s).

diffusion large language models (dLLMs) (Nie et al., 2025; Ye et al., 2025) predict and decode multiple positions over iterative denoising steps, offering a promising alternative with inherent parallelism. In fact, dLLMs typically adopt the global bidirectional attention during inference, which can misalign the decoding order with the causal structure of natural language, leading to unstable decoding trajectories and degraded generation quality. To mitigate this issue, a common practice is block-diffusion inference (Nie et al., 2025): the response is partitioned into predefined fixed blocks and decoded sequentially, so that the model follows an approximately left-to-right generation order while retaining parallel updates within each block, yielding substantial quality improvements across many tasks. More recent studies (Liu et al., 2025a; Google DeepMind, 2025; Labs et al., 2025; Bie et al., 2025) further investigate the scalability and optimization of dLLMs, advancing the generation of block-diffusion LLMs toward practical use.

Despite these advances, the fixed, predefined block schedule remains a key bottleneck (Lu et al., 2025). Figure 1 depicts an example adapted from (Lu et al., 2025), where the naive block scheduling requires the active block to be fully unmasked before decoding can proceed to the next block. An empirical verification is provided in Appendix A. This rigidity is agnostic to semantic information and difficulty: it can force the sampler to unmask positions even with quite low confidence (uncertain) inside the active block, while positions that are already high-confidence (certain) but lie just outside the active block cannot be decoded yet. Such premature commitments to uncertain positions tend to degrade generation quality, whereas delaying easy positions reduces parallelism and slows inference. Consequently, the fixed block scheduling yields a suboptimal quality–speed trade-off in semi-autoregressive dLLM inference.

These limitations suggest that an improved block schedule should account for semantic information and adapt to evolving context during denoising, rather than enforcing a static partition. Intuitively, we should wait on low-confidence positions inside the active region until they can benefit from more reliable context, and unmask high-confidence positions just outside the boundary earlier to avoid unnecessary delay. Existing attempts toward semantic-aware scheduling often rely on handcrafted heuristics or task-dependent cues (e.g., special separators or pattern-based rules), which may

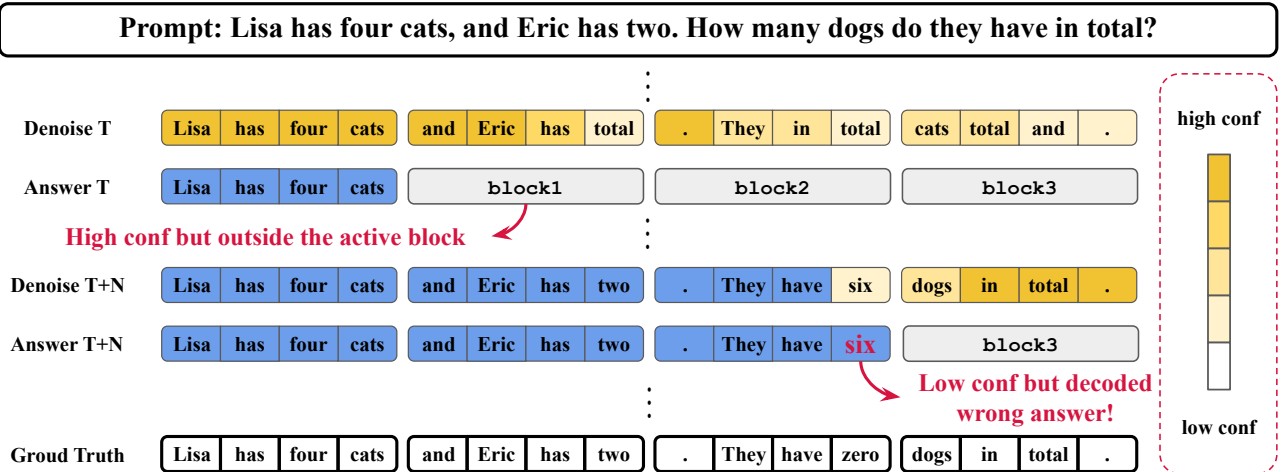

Figure 1. **Limitation of naive block scheduling.** At the denoising step T, several positions outside the active block have high confidence (dark yellow) but cannot be decoded due to the fixed block constraint, delaying easy tokens near the boundary. At later steps (T+N), the method is forced to decode low-confidence positions inside the active block, which can lead to premature, incorrect commitments (e.g., decoding "six" instead of the ground-truth "zero").

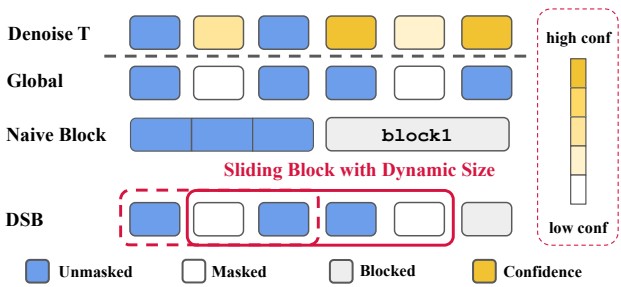

Figure 2. **A brief teaser of Dynamic Sliding Block (DSB).** At the denoising step T, the top row shows the confidence of masked positions (darker yellow indicates higher confidence). Global decodes the whole response simultaneously. Naive Block uses a fixed block, which can force early decoding of low-confidence positions inside the block and delay high-confidence positions outside it. In contrast, DSB employs a sliding block with dynamic size (red dashed box) to mitigate both issues.

not generalize across domains and model scales. This motivates a training-free scheduling mechanism that can adapt block boundaries online and consistently yield a better semi-autoregressive decoding strategy.

Based on this, we propose Dynamic Sliding Block (DSB), a training-free block scheduling for dLLMs (Figure 2). It maintains an active block whose size changes and boundary slides forward as decoding progresses. At each iteration, DSB updates the block based on the current unmasking state, expanding the active block whenever any position is unmasked and sliding forward once the positions adjacent to the block's start boundary are resolved. In this way, DSB alleviates the two key drawbacks of the naive block schedule: it avoids forcing premature commitments on low-confidence

positions within the active block, instead providing them with a more reliable context and unmasking high-confidence positions near the boundary in a timely manner, thereby improving both decoding quality and efficiency.

KV-cache is essential to fully exploit the efficiency gains brought by DSB. Directly applying existing training-free cache schemes for the naive block schedule (Ma et al., 2025; Wu et al., 2025b), e.g., prefix or dual caching with block-completion updates, to our sliding schedule can cause severe performance degradation. The reason is that when the block slides forward, it exposes new boundary positions, but the KV states are often still in flux because they or nearby tokens have just been updated, making them transient rather than stable (Ma et al., 2025). Treating such boundary tokens as long-lived cached states causes frequent invalidation and recomputation, offsetting the benefits of DSB.

We therefore propose DSB Cache, a training-free KV-cache mechanism tailored to DSB, aiming to improve efficiency while avoiding quality degradation. It introduces a prefix window immediately before the active block and refreshes KV states of both the prefix window and active block at each step, while caching remaining positions for reuse and performing periodic global refreshes. This prefix window stabilizes cache usage under block movement, substantially improves throughput and preserves generation quality.

Overall, DSB dynamically resizes and slides the active block to better adapt to the available context, and DSB Cache mitigates the KV-cache challenges introduced by block movement, yielding a stronger semi-autoregressive inference method. Extensive experiments across multiple models and benchmarks show that DSB, together with DSB

Cache, consistently improves the quality–speed trade-off.

Our contributions are summarized as follows:

- We propose **Dynamic Sliding Block (DSB)**, a training-free decoding schedule that improves dLLM performance as a stronger semi-autoregressive strategy. DSB analyzes and mitigates the limitations of the fixed, predefined block schedule, which ignores semantic information and difficulty, thereby improving both generation quality and inference efficiency.

- We introduce **DSB Cache**, a training-free KV-cache tailored to DSB, which can mitigate the severe performance degradation incurred by existing caching schemes. We identify and address cache instability caused by positions newly exposed as the active block slides, thereby further amplifying DSB's advantages.

- Extensive experiments across multiple models and benchmarks demonstrate the robust effectiveness of our method in improving the quality–speed trade-off over baselines. Comprehensive ablations further validate the effectiveness of each design and support our analysis.

## 2. Related Work

**Block-Diffusion LLMs.** Recent diffusion language models (Nie et al., 2025; Ye et al., 2025) have drawn increasing attention as an alternative paradigm for text generation, thanks to their ability to recover multiple tokens in parallel and incorporate bidirectional context during denoising. To improve generation quality, many systems (Nie et al., 2025; Wu et al., 2025b) adopt block-wise inference: directly applying a block-wise mask to those bidirectional dLLMs. This semi-autoregressive inference pattern better matches the causal feature of natural language and has been shown to substantially improve decoding stability in practice.

Beyond imposing block constraints only at inference time, a growing line of work (Wu et al., 2025a; Song et al., 2025; Bie et al., 2025; Liu et al., 2025a) trains models with block-causal masks to more closely match the block-diffusion LLMs' decoding schedule, yielding a paradigm that is causal across blocks yet parallel within blocks and aligning with KV-cache mechanisms. Fast-dLLM v2 (Wu et al., 2025a) proposes a data-efficient block diffusion framework that converts pretrained AR LLMs into block-wise diffusion decoders and accelerates inference via hierarchical caching and parallel decoding. LLaDA 2.0 (Bie et al., 2025) scales this training paradigm to 100B models and demonstrates industrially relevant performance, matching autoregressive models on multiple metrics and further validating the feasibility of dLLMs.

Given the limitations of fixed block schedules, several recent works have explored alternative block mechanisms (Lu et al., 2025; Yang et al., 2026; Liu et al., 2025a). AdaBlock-dLLM (Lu et al., 2025) identifies valid delimiters (such as "." and ",") and determines the range of the next block after the previous block is completely unmasked. Nevertheless, this delimiter-driven strategy may be less general when reliable delimiters are absent or incorrectly predicted. Wavefront-Diffusion (Yang et al., 2026) expands the positions near the already decoded token, while reducing the number of low-confidence positions. Therefore, discontinuous blocks may exist. As a concurrent work of ours, WeDLM (Liu et al., 2025a) is primarily motivated by the low efficiency of the current block diffusion KV Cache and its slower inference speed compared to AR, thus proposing a new training paradigm and an adaptive inference strategy based on the training model. In contrast, our method is motivated by the limitations of fixed block configurations and realizes a dynamically sized sliding block in a training-free manner, together with a tailored KV cache, leading to a more flexible and effective semi-autoregressive inference paradigm.

**Efficient Diffusion LLMs.** Despite dLLMs' ability to update multiple positions in parallel, practical inference remains challenging, motivating efforts on faster, more reliable decoding. Broadly, existing solutions fall into two directions: parallel decoding strategies (Wu et al., 2025b; Kim et al., 2025; Luo et al., 2026) and KV-cache mechanisms (Liu et al., 2025b; Ma et al., 2025; Wu et al., 2025b).

For parallel decoding (orthogonal to our method), Fast-dLLM (Wu et al., 2025b) leverages the positional dependence in dLLMs and introduces confidence-aware update rules to select which tokens to unmask in parallel. In contrast, EB-Sampler (Ben-Hamu et al., 2025) adopts an entropy-bounded criterion at each position to adaptively decide whether to commit that position at the current step. KLASS (Kim et al., 2025) uses token-level KL divergence across denoising steps to identify stable positions for parallel unmasking, improving decoding speed while largely preserving generation quality. Moreover, DAWN (Luo et al., 2026) estimates positional dependency relations from attention maps and uses them to schedule decoding more efficiently. Beyond these, other methods accelerate parallel decoding with more fine-grained scheduling strategies.

For KV caching, Fast-dLLM also introduces training-free cache designs for block-diffusion LLM inference, caching positions outside the active block, and periodically updating them to reduce redundant computation. dKV-Cache (Ma et al., 2025) further introduces delayed caching by caching tokens only after their KV states become stable, enabling near-lossless acceleration. Other approaches train with block-causal masks, yielding a natural lossless KV-cache strategy. Beyond the two main threads, techniques such as early stopping (Yang et al., 2025; Li et al., 2025) and distillation (Chen et al., 2025) have also been explored.

## 3. Background & Preliminaries

### 3.1. Diffusion LLMs Inference Procedure

Most recent diffusion LLMs follow the discrete masked diffusion model framework (Shi et al., 2025; Sahoo et al., 2024), formulating generation as an iterative unmasking procedure. In contrast to autoregressive models, dLLMs begin with a largely masked sequence and gradually fill in masked positions over multiple denoising steps until all [MASK] tokens are resolved. At each step, the model produces token distributions for the currently masked positions conditioned on the partially decoded sequence. Given a prompt $X$, we initialize the response as a fully masked sequence of predefined length $L$. In the traditional setting, the sampler unmasks one token with the highest confidence at each step. At each denoising step $t = 0, 1, \ldots, L - 1$, we input the concatenation of the prompt $X$ and the current response state $y^{(t)}$ to the model, and commit the token at the masked response position with the highest confidence:

$$i_t = \arg \max_{i \in M^{(t)}} \left( \max_{v \in \mathcal{V}} p_\theta \left( y_i = v \mid X, y^{(t)} \right) \right),$$

$$y_i^{(t+1)} = \begin{cases} \arg \max_{v \in \mathcal{V}} p_\theta \left( y_i = v \mid X, y^{(t)} \right), & \text{if } i = i_t, \\ y_i^{(t)}, & \text{otherwise.} \end{cases}$$

where $M^{(t)} \triangleq \{i \mid y_i^{(t)} = \text{[MASK]}\}$ is the set of masked response positions at step $t$, $\mathcal{V}$ is the vocabulary and [MASK] is the special mask token. Repeating this procedure for $L$ steps yields a fully unmasked response $y^{(L)}$.

However, when dLLMs decode many positions globally in parallel, the generation order can become misaligned with the causal structure of natural language: the model may prematurely commit to answer tokens before sufficient supporting context is formed. Such an order inversion often amplifies dependency conflicts and error propagation, resulting in degraded output quality. Given the inherently causal nature of language, practical dLLM inference frameworks therefore commonly adopt a block-wise decoding strategy. This semi-autoregressive paradigm substantially improves generation quality in current implementations.

### 3.2. Naive Block-diffusion LLMs

Recent advances leverage the block-causal structure of block diffusion to build more efficient KV caching schemes and high-throughput inference systems for dLLMs (Wu et al., 2025b;a; Bie et al., 2025; Liu et al., 2025a). In essence, block-diffusion LLMs inference is a constrained variant of standard dLLMs sampling, where a block-wise ordering is imposed on the unmasking process.

As shown in Figure 1, given a fixed response length $L$ and a block size $B$, one typically assumes that $B$ divides $L$ for simplicity. The response positions are then evenly partitioned into $K = L/B$ contiguous blocks, where $\mathcal{B}_k \triangleq \{(k-1)B + 1, \ldots, kB\}$ for $k = 1, \ldots, K$. At denoising step $t$, let the masked set be $M^{(t)} \triangleq \{i \mid y_i^{(t)} = \text{[MASK]}\}$, and the masked positions within block $k$ be $M_k^{(t)} \triangleq M^{(t)} \cap \mathcal{B}_k$. Decoding proceeds sequentially across blocks: the sampler operates on block $k$ and can move to block $k+1$ only when $M_k^{(t)} = \varnothing$. This constraint enforces global causality while retaining local parallelism, which typically yields higher generation quality in practice (Nie et al., 2025). However, the rigid block boundary may force premature commitment to low-confidence tokens within the active block, while delaying high-confidence tokens outside the block that could be safely decoded earlier, leading to suboptimal accuracy and efficiency (Lu et al., 2025).

## 4. Methodology

We propose Dynamic Sliding Block (DSB), a training-free decoding strategy to improve both accuracy and efficiency of diffusion large language models (Section 4.1). We further introduce DSB Cache, a KV-cache design tailored to DSB for more efficient inference (Section 4.2).

### 4.1. Dynamic Sliding Block

Naive block-diffusion LLMs (Nie et al., 2025; Wu et al., 2025b), while significantly improving generation quality over fully global parallel decoding, remain suboptimal in both quality and efficiency. The core issue (Lu et al., 2025) is that a predefined block schedule is independent of the actual semantics and difficulty of the sequence. It can force premature commitment to low-confidence positions inside the active block and delay high-confidence positions just outside the block boundary. Such a mismatch leads to suboptimal generation quality and decoding speed. We therefore propose Dynamic Sliding Block (DSB), which uses a sliding block with a dynamic size to incorporate semantic difficulty during inference. DSB preserves causality while enabling a better semi-autoregressive paradigm.

As shown in Algorithm 1, At iteration $t$, DSB maintains an active block $B^{(t)} = [s^{(t)}, e^{(t)})$ whose left and right boundaries advance at different rates. The block is initialized with size $S_{init}$. After each iteration, we update the left boundary by scanning the current block from left to right to find the first masked position. Then we set $s^{(t+1)}$ to the position immediately to its left. If the block contains no masked positions, we set $s^{(t+1)} = e^{(t)}$. For the right boundary, we expand the block to keep at least $S_{init}$ unresolved tokens whenever possible, while capping the block size at a maximum length $S_{max}$. Concretely, we set $e^{(t+1)} = \min(\ell_{prompt} + S_{init} + U, s^{(t+1)} + S_{max})$, where $\ell_{prompt}$ is the prompt length and $U$ is the number of totally unmasked positions. Under this rule, DSB maintains

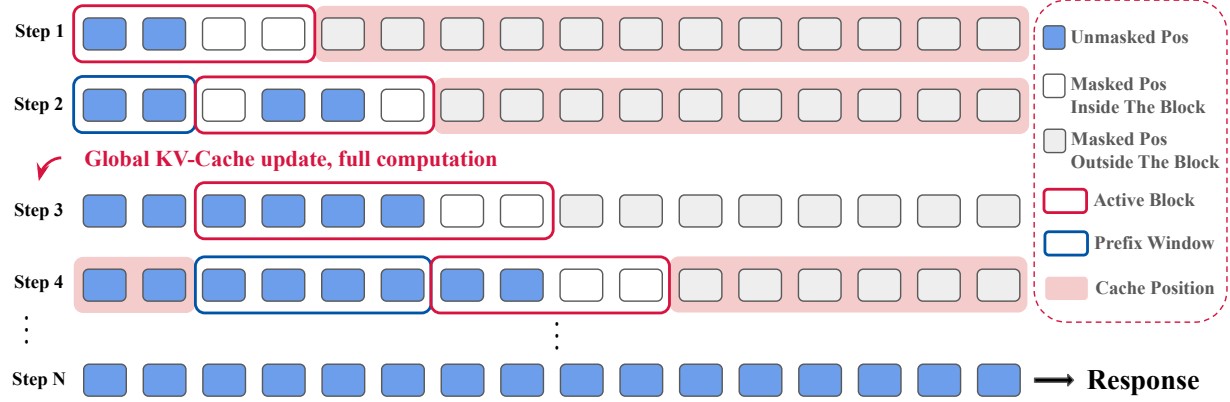

*Figure 3.* **Overview of DSB with DSB Cache.** DSB maintains an active block (red) that slides and can change its size across denoising steps, enabling globally causal yet locally parallel decoding. DSB Cache caches KV states for positions outside the active block (shaded region), while jointly refreshing the active block and the immediately preceding prefix window (blue) to handle boundary instability introduced by block movement. A periodic global cache refresh performs full computation to re-synchronize cached states.

---

**Algorithm 1** Dynamic Sliding Block (DSB) Inference

---

**Input:** prompt $X$ with length $\ell_{\text{prompt}}$, response length $L$, initial block size $S_{\text{init}}$, maximum block size $S_{\text{max}}$
Initialize $y \leftarrow ([\text{MASK}], \ldots, [\text{MASK}]) \in (\mathcal{V} \cup \{[\text{MASK}]\})^L$
Initialize $s \leftarrow \ell_{\text{prompt}}, e \leftarrow s + S_{\text{init}}$
Initialize decoded-token count $U \leftarrow 0$
**while** $U < L$ **do**
  $B \leftarrow \{s, \ldots, e-1\}$
  $M \leftarrow \{i \in B \mid y_i = [\text{MASK}]\}$
  Unmask a selected subset of tokens within $M$ and update $y$ accordingly.
  Update $U \leftarrow U + |\{i \in M \mid y_i \neq [\text{MASK}]\}|$
  Find $i^\star \leftarrow \min\{i \in B \mid y_i = [\text{MASK}]\}$ if it exists
  **if** $i^\star$ exists **then**
    Update $s \leftarrow i^\star$
  **else**
    Update $s \leftarrow e$
  **end if**
  Update $e \leftarrow \min(\ell_{\text{prompt}} + S_{init} + U, s + S_{max})$
**end while**
**Output:** decoded response $y$

---

local semantic coherence while adaptively decoding high-confidence positions earlier and postponing low-confidence positions until they can benefit from richer context, mitigating the limitations of the Naive block schedule and improving both quality and efficiency.

DSB recovers several useful cases. When $S_{max} = S_{init}$, the block can only slide without resizing, yielding a sliding block with a constant size (DSB (const.)). When $S_{max}$ is unbounded, DSB removes the upper bound on the block size, which can further improve throughput but weaken the causal constraint. We refer to this variant as DSB (greedy).

## 4.2. DSB Cache

KV caching is a critical technique for narrowing the inference-efficiency gap between diffusion LLMs and state-of-the-art autoregressive models. Prior work proposed training-free caching schemes for naive block-diffusion LLMs (Wu et al., 2025b), such as (i) prefix caching, which reuses KV states for positions before the current block, and (ii) dual caching, which reuses KV states for all positions outside the current block. Both schemes typically refresh cached states only when the current block is completed. However, directly applying these fixed-block cache update rules to DSB results in substantial quality degradation. To better match DSB, we propose DSB Cache, inspired by prefix and dual caching, while retaining both high throughput and output quality under dynamic block movement.

The key challenge is that DSB changes the active block boundaries over time. When the block slides, newly exposed positions have often just been decoded (either themselves or their nearby neighbors), and their KV states can be transient rather than stable (Ma et al., 2025). Treating such positions as long-lived cache entries causes frequent cache invalidation or expensive recomputation. At the same time, although dLLMs use full attention, the left-to-right nature of natural language makes inference rely more heavily on left-context positions in practice. Motivated by this, we design a prefix window update mechanism. We maintain an active window immediately before the current block and continuously refresh KV states within it to absorb boundary changes induced by sliding. Specifically, at step $t$, we set the prefix window length as

$$\ell_{pw}^{(t)} = \max(\ell_{pmin}, s^{(t)} - s^{(t-1)}),$$

*Table 1.* **Main results between DSB and baselines across LLaDA variants and five benchmarks.** We use Accuracy and TPS (tokens per second) as metrics to evaluate generation quality and efficiency.

| Method | | | GSM8K | | MATH | | HumanEval | | MBPP | | BBH | |
|---|---|---|---|---|---|---|---|---|---|---|---|---|
| Decode | Cache | Block | Acc.↑ | TPS↑ | Acc.↑ | TPS↑ | Acc.↑ | TPS↑ | Acc.↑ | TPS↑ | Acc.↑ | TPS↑ |
| LLaDA-8B-Instruct | | | | | | | | | | | | |
| Vanilla | None | Naive | 77.79 | 14.94 | 33.24 | 21.51 | 40.24 | 36.22 | 40.20 | 14.33 | 52.83 | 11.76 |
| Confidence | None | Naive | 77.26 | 48.70 | 33.02 | 56.73 | 40.85 | 119.1 | 40.40 | 52.72 | 52.77 | 53.27 |
| | | AdaBlock | 77.94 | 44.75 | 32.92 | 53.26 | 39.63 | 110.6 | 40.80 | 50.41 | 50.68 | 50.03 |
| | | DSB (const.) | 78.17 | 50.36 | **33.14** | **58.60** | **42.07** | **124.6** | 41.40 | 53.35 | 53.20 | 56.56 |
| | | DSB (greedy) | **78.54** | **51.03** | 33.06 | 58.02 | 39.63 | 123.8 | **41.80** | **53.56** | **53.60** | **57.21** |
| Confidence | Dual | Naive | 77.40 | 92.26 | 32.72 | 83.58 | 37.80 | 100.6 | 37.80 | 76.61 | 47.09 | 90.79 |
| | dKV | Naive | 77.56 | 59.36 | 33.04 | 58.94 | **40.85** | 89.97 | 40.80 | 54.61 | 49.38 | 60.71 |
| | Dual | AdaBlock | 78.17 | 80.42 | 32.70 | 77.15 | 39.02 | 96.01 | 40.20 | 72.83 | 45.02 | 85.56 |
| | DSB | DSB (const.) | 80.14 | 98.10 | 32.98 | 89.03 | 37.80 | 105.3 | **43.00** | **82.02** | 49.19 | 97.85 |
| | DSB | DSB (greedy) | **80.29** | **99.61** | **33.46** | **89.90** | 39.63 | **107.7** | 42.80 | 81.88 | **51.16** | **100.1** |
| LLaDA-1.5 | | | | | | | | | | | | |
| Vanilla | None | Naive | 80.52 | 14.05 | 33.50 | 19.29 | 44.50 | 12.78 | 39.20 | 5.320 | 57.44 | 12.11 |
| Confidence | None | Naive | 80.14 | 49.63 | 33.64 | 51.38 | 43.30 | 32.67 | **39.20** | **31.08** | 57.16 | 62.60 |
| | | AdaBlock | 81.04 | 43.82 | 33.10 | 49.15 | 40.24 | 31.00 | 37.80 | 25.77 | 56.50 | 57.81 |
| | | DSB (const.) | **81.20** | 48.60 | 34.00 | **51.80** | **43.30** | 32.71 | 36.60 | 30.58 | 58.53 | 66.16 |
| | | DSB (greedy) | 80.74 | **49.63** | **34.18** | 51.67 | 42.70 | **33.75** | 38.60 | 29.62 | **58.85** | **69.23** |
| Confidence | Dual | Naive | 80.82 | 86.42 | 32.36 | 74.59 | 34.10 | 30.91 | 34.20 | **50.61** | 55.08 | 108.7 |
| | dKV | Naive | 81.05 | 57.68 | 32.86 | 53.08 | 36.59 | 24.87 | **39.20** | 34.83 | 55.66 | 71.75 |
| | Dual | AdaBlock | 79.53 | 77.97 | 32.52 | 73.84 | 35.36 | 25.76 | 37.20 | 40.54 | 54.14 | 97.90 |
| | DSB | DSB (const.) | 81.73 | 93.99 | 33.90 | 78.70 | **35.98** | **32.40** | 34.00 | 48.04 | 57.47 | 115.7 |
| | DSB | DSB (greedy) | **81.96** | **95.54** | **33.96** | **78.94** | 34.70 | 31.92 | 37.00 | 47.61 | **58.06** | **120.1** |

where $\ell_{pmin}$ is the minimum window length. This ensures the window always covers the newly exposed region when the block advances while maintaining at least $\ell_{pmin}$ prefix tokens for stable reuse. This prevents unstable boundary tokens from being cached as long-lived prefix states.

DSB Cache combines prefix window updates with existing caching schemes in a simple manner: we cache (i) all positions outside the current active block and (ii) a prefix window immediately preceding the active block, which is refreshed together with the active block at every step to keep their KV states up to date. In addition, we periodically update the global cache after decoding $S_{init}$ tokens, amortizing the cost of cache maintenance and stabilizing cached states. This design is lightweight, and can effectively resolve the cache instability introduced by sliding blocks, enabling efficient inference with negligible quality loss.

## 5. Experiments

### 5.1. Configurations

**Models and Benchmarks.** We conduct the experiments on several benchmarks covering a range of reasoning, code generation, and general tasks: GSM8K (5-shot) (Cobbe et al., 2021), MATH (4-shot) (Hendrycks et al., 2021), HumanEval (0-shot) (Chen, 2021), MBPP (3-shot) (Austin et al., 2021), and BBH(3-shot) (Suzgun et al., 2023). Use variants of two models: LLaDA-8B-Instruct (Nie et al., 2025), LLaDA-1.5 (Zhu et al., 2025), Dream-v0-Base-7B (Ye et al., 2025), Dream-v0-Instruct-7B. We report the accuracy to reflect quality and tokens per second (TPS) to reflect speed.

**Baselines.** We compare our method against representative baselines along three axes: decoding strategy, block scheduling, and KV caching. For decoding, we consider the vanilla Top-1 sampling, which commits the single masked position with the highest confidence at each iteration, and the confidence-aware parallel decoding method from Fast-dLLM (Wu et al., 2025b), which commits multiple positions whose confidence exceeds a predefined threshold. For block scheduling, we adopt two baselines: Naive Block Scheduling, which decodes fixed, predefined blocks sequentially, and AdaBlock-dLLM (Lu et al., 2025), which identifies valid delimiters to adaptively determine the range of the next block. For KV caching, we adopt two baselines: Dual Cache, which stores KV states for all positions outside the active block and periodically synchronizes them during inference, and dKV-Cache-Decode (Ma et al., 2025), which delays KV caching until token representations become sufficiently stable across denoising steps.

*Table 2.* **Main results between DSB and baselines across Dream variants and five benchmarks.** We use Accuracy and TPS (tokens per second) as metrics to evaluate generation quality and efficiency.

| Method | | | GSM8K | | MATH | | HumanEval | | MBPP | | BBH | |
|---|---|---|---|---|---|---|---|---|---|---|---|---|
| Decode | Cache | Block | Acc.↑ | TPS↑ | Acc.↑ | TPS↑ | Acc.↑ | TPS↑ | Acc.↑ | TPS↑ | Acc.↑ | TPS↑ |
| | | | | | | Dream-v0-Base-7B | | | | | | |
| Vanilla | None | Naive | 75.06 | 21.09 | 34.76 | 28.10 | 48.78 | 49.77 | 53.00 | 29.01 | 51.39 | 23.33 |
| Confidence | None | Naive | 74.30 | **37.00** | 34.44 | 70.60 | 50.61 | **91.64** | 53.80 | **80.76** | 51.79 | 90.14 |
| | | AdaBlock | 74.30 | 31.62 | 34.50 | 61.05 | 50.61 | 83.90 | **54.20** | 67.57 | **52.56** | 73.59 |
| | | DSB (const.) | **74.83** | 34.16 | **34.70** | 72.05 | **50.61** | 91.41 | 54.00 | 80.30 | 51.76 | **91.80** |
| | | DSB (greedy) | 74.75 | 34.70 | 34.54 | **72.86** | 50.61 | 88.62 | 53.80 | 76.90 | 51.40 | 89.47 |
| Confidence | Dual | Naive | 73.31 | 68.04 | 33.28 | 92.30 | 52.44 | 74.49 | 53.00 | 101.1 | 50.30 | 127.3 |
| | dKV | Naive | 73.39 | 34.69 | 33.38 | 59.27 | 45.12 | 64.23 | 49.20 | 61.45 | 50.19 | 78.65 |
| | Dual | AdaBlock | 72.63 | 68.12 | 33.18 | 91.40 | 52.44 | 78.38 | 54.00 | 94.46 | 50.37 | 117.6 |
| | DSB | DSB (const.) | 74.22 | 68.71 | 33.52 | 96.44 | 52.44 | 78.09 | 54.80 | 115.3 | 50.01 | 140.1 |
| | DSB | DSB (greedy) | 72.48 | 68.09 | 36.94 | 92.99 | 51.83 | 73.11 | 55.00 | 108.5 | 49.95 | 132.6 |
| | | | | | | Dream-v0-Instruct-7B | | | | | | |
| Vanilla | None | Naive | 77.03 | 11.38 | 37.87 | 31.32 | 55.49 | 33.47 | 54.20 | 13.54 | 59.87 | 17.20 |
| Confidence | None | Naive | **75.51** | **45.79** | 38.28 | 60.59 | 60.37 | **77.61** | 55.40 | **62.03** | **58.62** | 96.07 |
| | | AdaBlock | 75.28 | 39.26 | **38.42** | 52.41 | 60.37 | 70.92 | 55.00 | 53.09 | 57.32 | 79.44 |
| | | DSB (const.) | 74.22 | 45.22 | 38.24 | **61.59** | 60.37 | 77.04 | 55.80 | 60.54 | 58.09 | **97.31** |
| | | DSB (greedy) | 75.06 | 38.52 | 38.36 | 60.63 | **62.20** | 74.58 | **56.00** | 55.67 | 58.13 | 91.88 |
| Confidence | Dual | Naive | 67.32 | 72.18 | 36.94 | 80.76 | 53.05 | 65.91 | 53.80 | 57.94 | 58.47 | 126.7 |
| | dKV | Naive | 71.87 | 42.12 | 37.02 | 49.87 | 56.10 | 53.70 | 51.80 | 47.45 | 55.89 | 83.56 |
| | Dual | AdaBlock | 67.02 | 69.67 | 36.90 | 79.56 | 49.39 | 64.60 | 54.40 | 63.16 | 56.29 | 118.3 |
| | DSB | DSB (const.) | 71.27 | 73.03 | 37.02 | 80.30 | 56.10 | 66.01 | 52.00 | 59.48 | 55.94 | 139.3 |
| | DSB | DSB (greedy) | 73.08 | 75.27 | 37.00 | 82.63 | 59.76 | 65.28 | 56.40 | 61.85 | 57.93 | 144.7 |

**Hardware and Implementation Details.** Our experiments were conducted on an `NVIDIA H200 140G` GPU. All evaluations are conducted using the official lm-eval (Gao et al., 2024) library. We set the generation length to 256 and the block length, as well as $S_{init}$, to 32 for the corresponding methods. The confidence threshold for parallel decoding is 0.9, according to the widely used official setting. $S_{max}$ has two settings: the block length (DSB(const.)) and unbounded (DSB(greedy)). The minimum window length $\ell_{pmin}$ is set to 24 for LLaDA series, and is set to 4 for Dream series.

## 5.2. Main Results

Table 1 and 2 report the accuracy and efficiency of DSB and the baselines under two scenarios: confidence-aware parallel decoding, and confidence-aware parallel decoding with KV caching, across four models and five benchmarks. Overall, DSB improves the quality–efficiency trade-off in most evaluated settings, while DSB Cache further strengthens this advantage and achieves consistent gains across models and benchmarks.

**Effectiveness of DSB with parallel decoding.** DSB exhibits stable and noticeable improvements on the LLaDA family under confidence-aware parallel decoding. Specifically, DSB (greedy) achieves 78.54 accuracy on GSM8K

with LLaDA-8B-Instruct, surpassing the vanilla sampling result, and achieves 51.03 TPS, indicating that DSB can deliver high quality while improving efficiency. Moreover, on HumanEval with LLaDA-8B-Instruct, DSB (const.) achieves 42.07 accuracy and 124.6 TPS, surpassing AdaBlock-dLLM in both accuracy and efficiency by 2.44 points and 14.0 TPS, respectively, which further demonstrates that DSB remains highly effective under a constant block size. In contrast, the gains on Dream are less consistent, likely because of its AR-initialized, shift-aligned training, which is discussed as a limitation in Section 6. Nevertheless, DSB still yields clear benefits in several settings; for instance, on MATH with Dream-v0-Base-7B, DSB (const.) achieves 34.70 accuracy with 72.05 TPS, improving both quality and efficiency under the same parallel decoding configuration. In addition, DSB (greedy) achieves outstanding accuracy on MBPP with Dream-v0-Instruct-7B, while incurring only minimal throughput loss. Overall, across most settings, DSB consistently delivers clear gains over naive block scheduling and AdaBlock-dLLM.

**Effectiveness of DSB with cache.** With DSB Cache, DSB delivers strong results across nearly all models and benchmarks, and in several cases even surpasses vanilla sampling in output quality. Specifically, on GSM8K, DSB (greedy) achieves 80.29 and 81.96 accuracy on the two LLaDA vari-

*Table 3.* **Ablation results on the effectiveness of DSB Cache.** We use two representative DSB variants (DSB (const.) and DSB (greedy)) on two benchmarks with LLaDA-8B-Instruct. We report Accuracy and TPS as measures of generation quality and efficiency. All settings are kept the same as in the main experiments.

| Method | | | GSM8K | | HumanEval | |
|---|---|---|---|---|---|---|
| Decode | Cache | Block | Acc.↑ | TPS↑ | Acc.↑ | TPS↑ |
| Vani | None | Naive | 77.79 | 14.94 | 40.24 | 36.22 |
| Conf | Dual | DSB(const.) | 76.42 | 78.93 | 28.05 | 87.79 |
| Conf | DSB | DSB(const.) | **80.14** | **98.10** | **37.80** | **105.3** |
| Conf | Dual | DSB(greedy) | 76.35 | 81.85 | 28.05 | 87.97 |
| Conf | DSB | DSB(greedy) | **80.29** | **99.61** | **39.63** | **107.7** |

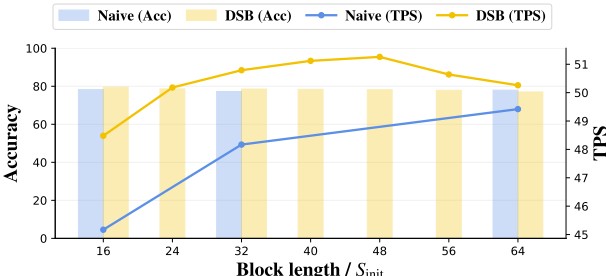

*Figure 4.* **Ablation results of block length under parallel decoding.** We compare DSB and naive block scheduling across different initial block lengths. Bars denote accuracy (left y-axis) and solid lines denote TPS (right y-axis). Results are reported on GSM8K with LLaDA-8B-Instruct.

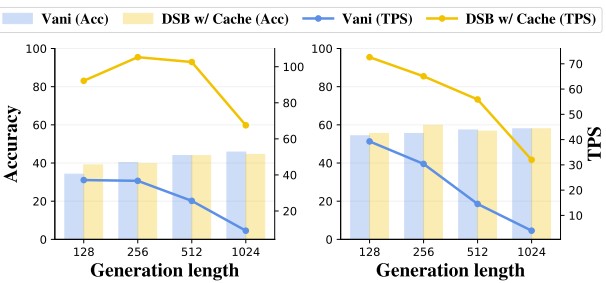

*Figure 5.* **Ablation results of generation length under parallel decoding.** We compare DSB with DSB Cache and the vanilla sampler across different generation lengths. Bars denote accuracy (left y-axis) and solid lines denote TPS (right y-axis). Results are reported on HumanEval, with the left and right panels corresponding to LLaDA-8B-Instruct and Dream-v0-Instruct-7B, respectively.

ants, both higher than vanilla sampling, while reaching 99.61 and 95.54 TPS, respectively—substantially above the 92.26 and 86.42 TPS of naive block with Dual Cache and other baselines. On Dream, DSB (const.) also performs competitively; for example, on HumanEval with the Dream base model, it attains 52.44 accuracy and 78.09 TPS. DSB (greedy) with cache also yields consistent gains on the Dream instruct model; on GSM8K, it achieves 73.08 accuracy and 75.27 TPS, outperforming Naive Block with dKV-Cache-Decode by 1.21 accuracy points and 33.15 TPS in this setting. Compared with the Naive Block with Dual Cache, Naive Block with dKV-Cache-Decode, and AdaBlock with Dual Cache, DSB with DSB Cache improves both quality and efficiency.

### 5.3. Ablation Results

We conduct ablation studies to assess the contributions of the prefix window in DSB Cache. We further examine the sensitivity of DSB to the block initial length, block max length, generation length, and the minimum window length $\ell_{pmin}$ to understand how these factors affect its effectiveness. All other experiment settings follow the main experiment.

**Effectiveness of DSB Cache.** Table 3 reports an ablation of DSB Cache on LLaDA-8B-Instruct under two representative DSB configurations, DSB (const.) and DSB (greedy). Overall, introducing the DSB Cache consistently improves both quality and efficiency on two benchmarks. In particular, removing the prefix window and directly applying Dual Cache leads to a substantial drop in both metrics across benchmarks and DSB settings. For DSB (const.), accuracy decreases from 80.14 to 76.42, and TPS drops from 98.10 to 78.93 on GSM8K. A similar degradation is observed for DSB (greedy) on both benchmarks, indicating that the prefix-window refresh in DSB Cache preserves essential KV states for sliding blocks and is critical for achieving strong quality and throughput with cached DSB inference.

**Effect of initial block length.** Figure 4 reports the accuracy and throughput of DSB and the naive block schedule under different initial block lengths $S_{init}$. Overall, DSB

consistently improves throughput across all tested settings and accuracy in most cases, except at $S_{init} = 64$. As $S_{init}$ increases, DSB largely preserves accuracy while its throughput stays within a stable range, typically rising at first and then slightly decreasing. When $S_{init}$ is overly aggressive, such as 64, the block reaches farther positions much earlier, weakening causality and slightly degrading performance. In comparison, naive blocks exhibit lower accuracy and increasing TPS as $S_{init}$ grows, yet remain below DSB in efficiency. Overall, DSB is more robust to the choice of initial block length and provides a better quality–speed trade-off.

**Effect of generation length.** Figure 5 reports the accuracy and throughput of DSB with DSB Cache under parallel decoding and the vanilla baseline with different generation lengths $L$. Across both LLaDA and Dream, DSB substantially improves throughput while maintaining or slightly improving accuracy in most cases, except on LLaDA at generation length 1024. As $L$ increases, throughput decreases for both methods due to higher denoising costs. Overall, it preserves a more favorable quality-speed trade-off across a wide range of lengths.

**Effect of max block length.** Figure 6 reports the evalua-

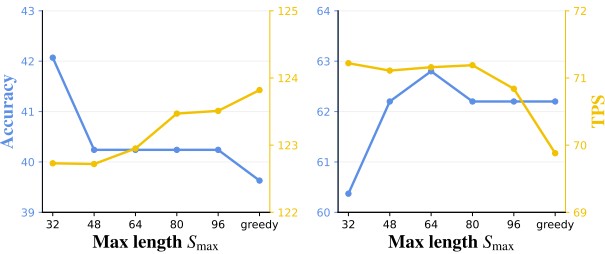

*Figure 6.* **Ablation results of** $S_{\max}$. We vary the maximum block length $S_{\max}$ and report accuracy (blue, left y-axis) and TPS (yellow, right y-axis) on HumanEval. The left and right panels correspond to LLaDA-8B-Instruct and Dream-v0-Instruct-7B, respectively.

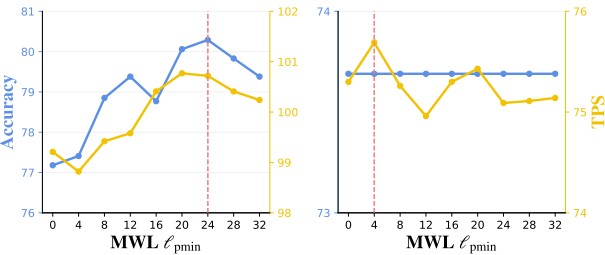

*Figure 7.* **Ablation results of** $\ell_{p\min}$ **with DSB Cache.** We vary the minimum prefix-window length $\ell_{p\min}$ and report accuracy (blue, left y-axis) and TPS (yellow, right y-axis) on GSM8K. The left and right panels correspond to LLaDA-8B-Instruct and Dream-v0-Instruct-7B, respectively. The red dashed line marks the default setting ($\ell_{p\min} = 24$ for LLaDA and $\ell_{p\min} = 4$ for Dream).

tion under different values of max block length $S_{max}$. We observe a clear quality–speed trade-off on LLaDA models. Increasing $S_{max}$ increases TPS by admitting more parallel updates, but can hurt accuracy due to less reliable low-confidence commitments. Conversely, Dream exhibits a different trend: TPS gradually decreases as $S_{max}$ increases, while accuracy follows a rise-then-fall pattern, with $S_{max} = 64$ serving as a clear turning point.

**Effect of minimum window length.** Figure 7 reports the evaluation under different values of minimum window length $\ell_{pmin}$. We observe that as the minimum window length $\ell_{pmin}$ increases, the LLaDA setting exhibits an overall rise-and-fall trend in both accuracy and throughput, with $\ell_{pmin}{=}24$ as a clear turning point. In contrast, for Dream, accuracy decreases slightly, and throughput generally decreases as $\ell_{pmin}$ increases. The default $\ell_{pmin} = 24$ and $\ell_{pmin} = 4$ (dashed line) maintains a high generation quality and improved efficiency.

## 6. Limitations

Const. and Greedy are two simple, practical DSB variants that offer clear implementation choices while maintaining robust empirical performance. In fact, as shown in Figure 6,

the effect of the maximum block length varies slightly across datasets and even across model architectures. Since this factor has a relatively small impact in both our experiments and theoretical analysis, we believe that the simpler Const. and Greedy settings, which exhibit clearer trends, are more general and practical.

While DSB demonstrates superiority in most cases, its performance on the Dream model without a KV cache is somewhat inconsistent. Nevertheless, under the most widely used setting: inference with a KV cache, DSB still exhibits relatively consistent superiority, improving both accuracy and efficiency.

## 7. Conclusion

In this work, we improve the performance of diffusion LLMs in both generation quality and inference speed by developing a stronger semi-autoregressive inference strategy. Specifically, we propose Dynamic Sliding Block (DSB), a training-free block schedule for dLLMs that maintains a moving block with a dynamic size, mitigating the limitations of fixed, predefined block schedules that ignore semantic information. Moreover, we introduce DSB Cache to enable efficient inference over sliding blocks, further amplifying DSB's practical advantages. Together, DSB and DSB Cache offer a new perspective on block scheduling for dLLMs and raise the achievable quality–speed frontier. Extensive experiments across multiple models and benchmarks demonstrate the effectiveness and robustness of our approach.

## Impact Statement

This paper presents work whose goal is to advance the field of machine learning. There are many potential societal consequences of our work, none of which we feel must be specifically highlighted here.

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

## A. An Empirical Verification of Motivation

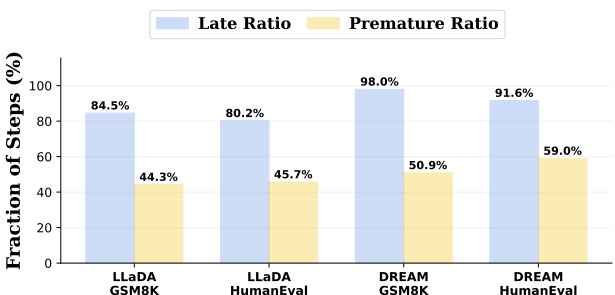

Figure 8. **An Empirical Verification of Motivation.** We collect the frequencies of late decoding and premature decoding on GSM8K and HumanEval with LLaDA-8B-Instruct and Dream-v0-Instruct-7B.

Our work follows AdaBlock's (Lu et al., 2025) key observations, and the relevant proofs are already presented in that paper. To further demonstrate the reliability of this motivation, we conducted a lightweight verification on LLaDA-8B-Instruct and DREAM-v0-Instruct-7B, counting the frequencies of **late steps** (delaying easier positions outside the active block) and **premature steps** (forcing the commitment of uncertain positions within the block). As shown in Figure 8, late decoding occurs in over 80% of steps and premature decoding in 44–59% of steps across both models and benchmarks, which fully proves that this phenomenon does exist.

**Implementation details.** We use 50 samples each from GSM8K and HumanEval; other settings are the same as the main experiment. At every denoising step, we compute the confidence of all masked positions and count two events: a **late step**, where at least one masked position *outside* the active block already has confidence $\geq 0.9$ but is held back from being committed; and a **premature step**, where at least one position *inside* the block is unmasked with confidence $< 0.9$. Late Ratio = Late Step/Total Step, and the same applies to the premature step.

## B. Effect of Suffix Window

Inspired by the strong performance of the prefix window, we further explore a symmetric design that maintains a suffix window after the active block under cached decoding. Unlike the prefix window, which must cover newly exposed positions induced by block sliding, the suffix window simply keeps a fixed-length region immediately following the active block and refreshes the KV states of these suffix positions on-the-fly. This design aims to provide fresher representations for upcoming tokens and potentially improve cached inference with sliding blocks.

Figure 9 reports the evaluation under different values of suffix window length. We find that, as the suffix window length

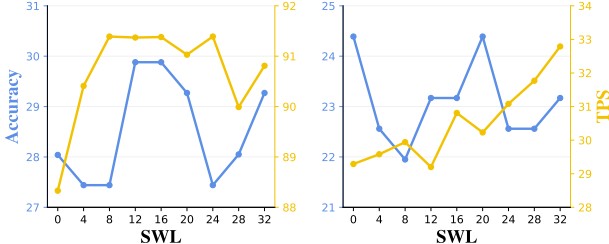

Figure 9. **Results of suffix window length (SWL) with DSB Cache.** We vary the suffix window length and report accuracy (blue, left y-axis) and TPS (yellow, right y-axis) on HumanEval. The left and right panels correspond to LLaDA-8B-Instruct and LLaDA-1.5, respectively.

increases, LLaDA-8B-Instruct shows a clear rise-and-fall pattern in both accuracy and throughput, with moderate window lengths yielding relatively strong performance. However, the suffix window is much less effective on LLaDA-1.5: while throughput generally increases with larger SWL, accuracy consistently stays below the SWL = 0 baseline, providing no clear quality benefit. Overall, unlike the prefix window, the suffix window does not demonstrate consistent advantages, and we therefore do not adopt this design in our final method.

## C. Discussion and Future Work

Block-masked dLLMs are becoming an increasingly common design choice, as this semi-autoregressive inference paradigm preserves the left-to-right causality of natural language while still enabling parallel updates within each step. However, the widely used fixed, predefined block schedule is agnostic to semantic difficulty and therefore remains a suboptimal strategy in practice. To better balance causality and parallelism, we propose Dynamic Sliding Block (DSB), and show both theoretically and empirically that it provides a stronger semi-autoregressive scheduling strategy, addressing the key limitations of naive fixed-block decoding from multiple perspectives.

As block-diffusion LLMs based on fixed blocks continue to mature, we hope DSB can serve as a more effective alternative that further advances dLLM inference. We also anticipate future work that builds on DSB, including additional inference optimizations and training approaches that incorporate DSB-like schedules during pretraining or post-training to better align model behavior with dynamic block scheduling.

