# OpenReview forum: "DSB: Dynamic Sliding Block Scheduling for Diffusion LLMs"
_ICML.cc/2026/Conference — ICML 2026 regular_

### Official Review · Reviewer_TWPe · 2026-03-12

**Soundness:** 2
**Presentation:** 3
**Significance:** 3
**Originality:** 2
**Overall Recommendation:** 4
**Confidence:** 3

**Summary:**

This paper identifies the rigidity of traditional fixed-size block methods in discrete diffusion models, which often forces the model to unmask low-confidence tokens. To address this, the authors propose Dynamic Sliding Block (DSB), a training-free strategy where the active block's boundaries evolve at different velocities on two sides to enhance sampling flexibility. Moreover, to improve sampling efficiency, the authors introduce DSB cache, which additionally updates a prefix window before the active block to resolve KV-state instability caused by boundary movement. The authors conduct experienments on math and code tasks with LLaDA and Dream models.

**Compliance With Llm Reviewing Policy:**

Affirmed.

**Final Justification:**

I thank the authors for the detailed clarification. The follow-up comments addreassed most of my concerns. Thus, I will increase my score to 4.

However, the marginal and inconsistent performance gains remain a primary concern. The final manuscript still requires a more detailed analysis of the inconsistent improvements on the Dream model and a clearer description of baseline settings.

**Key Questions For Authors:**

Please see the weakness.

**Limitations:**

Yes

**Strengths And Weaknesses:**

## Strengths
1. Soundness: The authors conduct extensive experiments across 5 datasets using 2 models to verify the effectiveness.
2. Presentation: The paper is well structured, with the core motivation and the proposed method clearly presented.
2. Significance: Optimizing the sampling schedule for discrete diffusion models is an important research problem. The proposed method provides value for practical deployment.


## Weakness
1. Soundness:
    - The motivation is not well-supported by empirical evidence. Although it is intuitive that a fixed block size limits sampling flexibility and may force the decoding of low-confidence tokens, the argument would be more convincing if the authors could provide statistical data to quantify these occurrences.
    - Performance gains are inconsistent across settings. For example, DSB fails to surpass baselines in several configurations with Dream models. Additionally, the method appears highly sensitive to the choice of hyperparameters.
    - About the practical benefit of DSB Cache: Figure 7 shows that DSB Cache does not improve the accuracy of the Dream-v0-Instruct-7B model.
    - It is better to provide comparison with more baselines or simpler alternatives, for example, setting the right boundary of the active block to the position of the farthest unmasked token plus a fixed offset, or existing method like AdaBlock [1].
2. Originality:
    - The concept of eliminating fixed block schedules has been explored in some existing literature [1,2]. Since similar ideas have been proposed in prior work, the authors should explicitly discuss the technical distinctions between their proposed method and these existing approaches.

[1] Lu et al., Adablock-dllm: Semantic-aware diffusion llm inference via adaptive block size, ICLR 26.

[2] Yang et al., WavefrontDiffusion: Dynamic Decoding Schedule for Improved Reasoning, ICLR 26.

---

> ### Author Rebuttal · Authors · 2026-03-31
>
> Thanks for the insightful and careful comments. All review comments will be taken into consideration in the next revision.
> ### 1. The empirical evidence of motivation.
> Thanks for your suggestions. Our work follows AdaBlock's key observations, and the relevant proofs have already been explained in this paper. To further demonstrate the reliability of this motivation, we conducted a lightweight verification. We statistically analyzed the step frequencies of late decoding (delaying the decoding of easier positions outside the block) and premature decoding (forcing the commitment of uncertain positions within the block). A denoising step with confidence > 0.9 outside the block is considered a late step, while a decoding confidence below 0.5 is considered a premature step. $LateRatio = LateStep / TotalStep$. The same applies to the premature ratio. The result is at https://ibb.co/0y51v9SP. This figure illustrates that this phenomenon of late decoding and premature is frequently observed in code and math tasks, which validates our motivation.
>
> ### 2. Performance Inconsistency and sensitivity of hyperparameter.
> (1). Performance Inconsistency. Due to the inherent differences in the mechanisms of the two model series, DSB exhibits somewhat inconsistent results on the Dream series model, especially when no KV cache is used. This should be considered a limitation.
>
> (2). Sensitivity of hyperparameter. We analyzed the impact of the hyperparameters $S_{max}$ and $l_{pmin}$ and found that the results exhibit a clear trend trade-off within a narrow range. Considering the acceptable impact, for simplicity and universality, we introduced two modes for $S_{max}$: const and greedy. Therefore, only $l_{pmin}$ is currently used as the hyperparameter (a detailed explanation of this hyperparameter is in answer 3). Figure 7 shows that this introduces some fluctuations, and we selected the better result based on experimental results to find the most reasonable settings.
>
> ### 3. DSB Cache and Figure 7.
> Thank you for your careful review. The term $l_{pmin}$, referring to the minimal prefix window length, may cause confusion with the prefix window length ($l_{pw}$). In fact, $l_{pw} = max(l_{pmin}, s^{t}-s^{t-1})$, and the prefix window exists regardless of how $l_{pmin}$ is set. Therefore, the DSB Cache actually contributes to each data point. Apologies for this ambiguous parameter name; the original purpose of Figure 7 was to analyze the impact of $l_{pmin}$.
>
> ### 4. Comparison with more baselines
> Thanks again for your suggestions.
>
> The comparison result table is at https://ibb.co/7xGyh0yn. All experiments were conducted under the same setting as the main experiment; the full benchmark evaluation will be added in the next revision. From the table, the accuracy appears correct, but the TPS is abnormal. After investigation, the machine's CPU is the problem. Suspicious results are marked in red. **I will address this issue as soon as possible and report the accurate data during the subsequent rebuttal stage.** Sorry for that, and thank you for your understanding.
>
> However, from an accuracy perspective, DSB still outperforms AdaBlock in most cases, except for a Dream model without a KV cache. In particular, DSB with Cache outperforms AdaBlock by almost 8 in the Dream model and HumanEval task.
>
> ### 5. Distinction from existing works.
> (1). Distinction from AdaBLock. (i). Block scheduling algorithm. AdaBlock identifies valid delimiters (such as "." and ",") and determines the range of the next block by performing an additional forward after the previous block is completely unmasked. However, DSB uses dynamic block scheduling, adjusting the number of blocks after each forward based on the number of decoded blocks in that forward. (ii). KV Cache. AdaBlock's KV cache implementation directly applies the fast-dllm KV cache method. In contrast, DSB Cache is a caching method designed for DSB itself, maintaining high quality by maintaining a prefix window.
>
> (2). Distinction from WavefrontDiffusion. (i). Block scheduling algorithm. WavefrontDiffusion employs an "Expand" followed by "Prune" approach. "Expand" expands the positions near the already decoded token, while "Prune" reduces the number of low-confidence positions. Therefore, discontinuous blocks may exist. However, DSB expands forward depending on the number of decoded blocks in that forward. Then the right boundary moves the corresponding distance. (ii). KV Cache. WavefrontDiffusion did not design a KV cache specifically, nor did it include any experiments utilizing existing KV caches. However, KV caches are a key mechanism for improving performance. DSB, on the other hand, proposed optimizations specifically for KV caches, namely DSB Cache.

---

> > ### Author Rebuttal · Reviewer_TWPe · 2026-04-04
> >
> > I thank the authors for their detailed responses. While the rebuttal addresses some of my concerns, my main concern about the effectiveness of the proposed method still remains.
> >
> > - As the authors mentioned, the method does not achieve consistent improvements on the Dream model.
> >
> > - Regarding the baseline method AdaBlock:
> >    - There are still some issues with the reported TPS results.
> >    - There is a discrepancy between the results reported by the authors and those reported in the original AdaBlock paper. Compared to the vanilla method, the results reported in the AdaBlock paper show a more significant improvement. I would like to kindly request an explanation for this. If the results from the AdaBlock paper are used, the improvement of DSB over AdaBlock is marginal.

---

> > > ### Author Response · Authors · 2026-04-04
> > >
> > > Thank you very much for your time and your valuable comments. I also appreciate your understanding regarding the hardware issue. I have now repeated the experiment on another proper H200 machine. All of your suggestions will be taken into account in the next revision. I hope that the following responses fully address your concerns.
> > >
> > > ### 1. DSB does not consistently improve performance on the Dream model without the KV Cache.
> > >
> > > Thanks for your valuable comment. I will discuss this issue in detail in the limitations section of the next revision. While DSB demonstrates superiority in most cases, its performance on the Dream model without a KV cache is somewhat inconsistent. Nevertheless, under the most widely used setting: inference with a KV cache, DSB still exhibits relatively consistent superiority, improving both accuracy and efficiency.
> > >
> > > ### 2. Correction of previous TPS results.
> > >
> > > Thank you again for your previous understanding. I need to correct the inaccuracies of TPS (due to the hardware issues) in my previous answer. I have repeated the experiments on another properly configured H200 machine.
> > >
> > > Correction of the comparison with AdaBlock. The result table is at https://ibb.co/ZpjGtG3h. Overall, DSB outperforms AdaBlock in both accuracy and TPS, except for a slight underperformance on the Dream model without a KV cache. Specifically, DSB achieves an improvement of nearly 8 accuracy points on the HumanEval task for the Dream model with KV Cache, and an improvement of almost 22 TPS on the GSM8K task for the LLaDA model with KV Cache.
> > >
> > > ### 3. Discrepant results on AdaBlock-dLLM are mainly caused by different settings (the generation length is 512 in AdaBlock original paper but 256 in our current experiments).
> > >
> > > Thank you for your careful review. **The main reason for this discrepancy is the different generation length settings.** I checked the AdaBlock paper and found that our experimental conditions were not the same: **the generation length in AdaBlock original paper is 512, but in DSB main experiment settings it is 256. All experiments in the rebuttal follow the main DSB experimental settings.** Therefore, a difference in the results is expected. Furthermore, comparing previous DLLM related papers, I also observed discrepancies in the experimental results between AdaBlock and Fast-dLLM. Under the same experimental settings, AdaBlock: vanilla is 76.7, dynamic is 77.6; Fast-dLLM: vanilla is 77.5, dynamic is 77.6. These differences are likely due to variations in experimental software environments, and such discrepancies are common in practice.
> > >
> > > Meanwhile, the current experiments were conducted directly using AdaBlock's open-source code in my current environment. I checked the accuracy for two runs during rebuttal (different H200 machines but the same Docker images), and the accuracy was identical. I also double-checked my experimental code, confirming the reliability of the results.
> > >
> > > I apologize for any misunderstanding caused by the discrepancy in the results, which is mainly due to different experimental settings (DSB's generation length = 256 and AdaBlock's (original paper) generation length = 512), and may also be slightly influenced by differences in software environments. I can confirm that the experimental results are correct.
> > >
> > > Thanks again for your careful review. I hope this fully addresses your concerns. Please feel free to ask any further questions.

---

### Official Review · Reviewer_9L5V · 2026-03-12

**Soundness:** 2
**Presentation:** 3
**Significance:** 3
**Originality:** 3
**Overall Recommendation:** 5
**Confidence:** 3

**Summary:**

Authors in this paper proposed training-free block schedule for Dynamic-Sliding block (DSB) for diffusion Large Language Models (dLLMs) to address semantic information ignored by fixed block schedules. The proposed idea respects the natural causality that exists in
language, by giving the flexibility of adapting block size and also the boundaries of the block, depending on the tokens decoded with high confidence.This improved the generation quality.
Moreover, they proposed a DSB cache mechanism that handles KV-cache in dynamically sliding boundaries where existing techniques like dual cache and prefill cache fail. This amplified the inference speed.
They evaluated their proposed approach using two variants of dLLMs : LLaDA-8B-Instruct,LLaDA-1.5, Dream-v0-Base-7B, Dream-v0 Instruct-7Bon on benchmarks like GSM8K, MATH, HumanEval, MBPP, and BBH. In most of the experiments, DSB with DSB cache achieves improved accuracy and inference speed.

**Compliance With Llm Reviewing Policy:**

Affirmed.

**Final Justification:**

Thanks for the authors' response to my questions. They addressed my questions about comparison with Adablock with experimental results and also ablation studies with dynamic sliding block vs dynamic size. Hence, I am increasing my score.

**Key Questions For Authors:**

1. Including missing data points in figure 4 for block length for the naive approach will help solidify the consistency claim and strengthen the paper
2. Could rectify the explanation for accuracy for figure 5 in line number 421 to reflect the result.
3. Accuracy and TPS reported in Table 1 shows that DSB with DSB cache is definitely better but compared with const and greedy approaches. Where Figure 6 shows that greedy is not the best, so, the results might be even better. A comment on this one in the paper would strengthen the paper.
4. Since Adablock-dLLM is discussed in background section. It would make the paper very good if discussion or comparison with Adablock-dLLM in theory or experiments to see how sliding boundaries will help with language semantics.  As Adablock-dLLM also has adaptive block sizes, a discussion on how well dynamic boundaries help generational quality better.
5. a small minor grammatical correction : line number 83 is repeated from 79

**Limitations:**

While the paper discusses suffix window limitations (Appendix A), it does not explicitly frame the causality-speed trade-offs shown in the ablations in Figure 6 as limitations. The model-dependent behavior (Dream vs. LLaDA showing different patterns) suggests DSB's applicability varies by architecture, but this is not discussed as a limitation. A dedicated limitations section addressing when DSB is most beneficial would strengthen the paper.

**Strengths And Weaknesses:**

Strengths:
Proposed DSB idea is novel and very interesting, handling and mitigating the current mismatch between the language semantics and the existing dLLMs that won’t decode high confidence tokens outside the active block.
Related work section is thorough and well presented.
The Dynamic Sliding Block technique is well explained in Section 4 with Figure 3. And in the introduction, Section 1, Figure 1 and Figure 2 give a very good explanation of why existing sliding block techniques fail intuitively.
Datasets used in experiments are well done, covering reasoning, coding, and general tasks


Weaknesses:
Missing points in the Figure 4: In figure 4, ablation studies for block length results for naive approach are missing for lengths = 24,40,48, 56. And it looks like Naive accuracy is
better than DBS accuracy for block length=64, which is inconsistent to the stated explanation
in the line number: 373, right column.
Paper discusses AdaBlock-dLLM in the related work section as the closest. Baseline comparisons in experiments are not compared with AdaBlock-dLLM technique or discussed theoretically.
In Figure 5, the text (line 421) states DSB ‘consistently maintains or even slightly improves
accuracy over the vanilla sampler.’ However, the figure shows vanilla outperforming DSB on
accuracy for generation length=1024 on LLaDA. The text should be revised to acknowledge
this trade-off, while noting that DSB provides substantial TPS improvements.
Table 1 reports only two DSB variants (const and greedy). While, Figure 6 shows that greedy
is definitely not the best setting for accuracy.

---

> ### Author Rebuttal · Authors · 2026-03-31
>
> Thanks for the insightful and careful comments. All review comments will be taken into consideration in the next revision.
> ### 1. Missing data points and the inconsistent explanation of Figure 4.
> (1). Missing data points. Because the predefined fixed block requires dividing the generated length, it is difficult to achieve block lengths that are not powers of 2, such as 24 or 28, while maintaining the general implementation.
>
> (2). Inconsistent explanation. Thank you very much for your suggestion. I will show an accurate description of the results in the graph in the next revision. DSB experiences performance degradation when the initial block size is extremely large; this phenomenon is briefly analyzed in lines 376-378, but the result description is not precise enough. Although DSB can provide an efficiency advantage in this case, it comes at the cost of accuracy. Thank you for your careful review.
>
> ### 2. Rectification of Figure 5.
> As shown in Figure 5, there is a slight loss of accuracy in the LLada section length, although it maintains an efficiency advantage. Thank you again for your careful review.
>
> ### 3. Results could be better.
> Const. and Greedy are not necessarily the two optimal strategies. In fact, we have observed that the effect of the max block length varies slightly across datasets and even models. Because the effect has a relatively small impact in experiments and theory, we believe that the simple settings of const and greedy, which have a clear trend, are more general.
>
> ### 4. Distinction and comparison of AdaBlock-dLLM
> (1). Distinction from AdaBLock. (i). Block scheduling algorithm. AdaBlock identifies valid delimiters (such as "." and ",") and determines the range of the next block by performing an additional forward after the previous block is completely unmasked. However, DSB uses dynamic block scheduling, adjusting the number of blocks after each forward based on the number of decoded blocks in that forward. (ii). KV Cache. AdaBlock's KV cache implementation directly applies the fast-dllm KV cache method. In contrast, DSB Cache is a caching method designed for DSB itself, maintaining high quality by maintaining a prefix window.
>
> (2). The comparison result table is at https://ibb.co/7xGyh0yn. All experiments were conducted under the same setting as the main experiment; the full benchmark evaluation will be added in the next revision. From the table, the accuracy appears correct, but the TPS is abnormal. After investigation, the machine's CPU is the problem. Suspicious results are marked in red. **I will address this issue as soon as possible and report the accurate data during the subsequent rebuttal stage.** Sorry for that, and thank you for your understanding.
>
> However, from an accuracy perspective, DSB still outperforms AdaBlock in most cases, except in the case of a Dream model without a KV cache. In particular, DSB with Cache outperforms AdaBlock by almost 8 in the Dream model and Humaneval task.
>
> ### 5. Thanks for the advice on grammar correction and your careful review.
>
> ### 6. A limitations section.
> (1). Figure 6. Figure 6 shows that the hyperparameter $S_{max}$ yields a narrow range of quality-speed trade-offs. Considering the simplicity and limited impact on the results, we chose both Const and Greedy settings. This process should be considered a limitation.
>
> (2). Performance applicability across models. Due to the inherent differences in the mechanisms of the two model series, DSB exhibits somewhat inconsistent results on the Dream series model, especially when no KV cache is used. This should be considered a limitation.

---

> > ### Author Rebuttal · Reviewer_9L5V · 2026-04-03
> >
> > I thank the authors for the detailed explanations and additional results addressing my concerns. I can now better appreciate the advantages of the proposed Dynamic Sliding Block approach over AdaBlock through the provided experiments. However, the underlying mechanism behind this improvement is still not entirely clear to me, specifically where the advantage is coming from, is it from the sliding block or the dynamic size. However, additional experiments in the rebuttal does prove that dynamic sizing helps empirically.
> >
> >
> > Overall, I appreciate the additional results compared with Adablock and am inclined to slightly increase my score.

---

> > > ### Author Response · Authors · 2026-04-04
> > >
> > > Thank you very much for your time and your willingness to raise the score. I also appreciate your understanding regarding the hardware issue. I have now repeated the experiment on another proper H200 machine. All of your suggestions will be taken into account in the next revision. I hope that the following responses fully address your concerns.
> > >
> > > ### 1. Correction of previous comparisons.
> > >
> > > First, I need to correct the inaccuracies of TPS (due to the hardware issues) in my previous answer. I have repeated the experiments on another properly configured H200 machine. Thank you again for your previous understanding.
> > >
> > > Correction of the comparison with AdaBlock. The result table is at https://ibb.co/ZpjGtG3h. Overall, DSB outperforms AdaBlock in both accuracy and TPS, except for a slight underperformance on the Dream model without a KV cache. Specifically, DSB achieves an improvement of nearly 8 accuracy points on the HumanEval task for the Dream model with KV Cache, and an improvement of almost 22 TPS on the GSM8K task for the LLaDA model with KV Cache.
> > >
> > > ### 2. **Contributions of Sliding Block and Dynamic Size**.
> > >
> > > (1). Theoritical discussion. The original intention was to consider the Sliding Block and Dynamic Size as a unified design. As inllustrated in figure at https://ibb.co/0p4S5VHS, regarding the left and right boundaries of a block, **Dynamic Size** only controls the movement of the right boundary, which is determined by the number of decoded elements in the current forward pass. Therefore, the block size will continuously increase, which is detrimental to the efficiency of KV Cache. **Sliding Block** focuses on the movement of the left boundary: it moves the left boundary to the first undecoded position at each step. If there is only a Sliding Block, then the right boundary simply moves along with the left boundary (DSB (const.)). If we only apply Sliding Block, the right boundary simply moves together with the left boundary (i.e., DSB (const.)). In this case, as shown in the figure, the 5th position cannot be decoded at step 2 under Sliding Block, which also leads to efficiency issues. Thus, they are designed to work together, and Sliding Block can be viewed as the special case of DSB with a constant block size (DSB (const.)).
> > >
> > > (2). Experical discussion.But to better measure their contributions, I conducted a simple ablation study on the LLaDA model on the proper H200 server. We compared three variants: Dynamic Size, Sliding Block (DSB (const.)) and DSB (greedy.), where DSB (greedy) can be seen as a combination of Sliding Block and Dynamic Size. All methods use the confidence-based parallel decoding algorithm.
> > >
> > > The result table is provided at https://ibb.co/ZRj3sRdV. Overall, these two submodules do not offer any particularly significant advantages, except that the Sliding Block tends to contribute more efficiency when using the KV Cache. DSB (greedy) achieves a relatively balanced advantage in both accuracy and TPS by leveraging the simultaneous effect of the two modules. Specifically, with DSB and DSB Cache, DSB achieves an accuracy of 80.29 on GSM8K and 116.4 TPS on HumanEval, both of which are the best among the three variants. The remaining metrics are also better than those of the worst-performing submodule. In the case of DSB without cache, Dynamic Size is essentially the same as DSB (greedy) because the full attention needs to be calculated in each forward computation, resulting in the same computational cost. Furthermore, because their right boundary movement patterns are the same, the content decoded for position is identical. Overall, they perform similarly. For the remaining results, it seems that both Dynamic Size and Sliding Block will introduce some degree of fluctuation across different tasks, and finally bring DSB to a balance even better level.
> > >
> > > Considering both the theoretical analysis and the experimental results, we are inclined to regard Sliding Block and Dynamic Size as a single integrated mechanism, DSB, while treating the variant with only Sliding Block but without Dynamic Size as a special case of DSB (DSB (const.)). Together, Sliding Block and Dynamic Size implement different treatments of the left and right boundaries of a block, jointly forming the overall DSB design.
> > >
> > > **Thanks again for your willingness to raise the score. I hope this fully addresses your concerns. Please feel free to ask any further questions.**

---

### Official Review · Reviewer_C9bf · 2026-03-23

**Soundness:** 2
**Presentation:** 3
**Significance:** 3
**Originality:** 2
**Overall Recommendation:** 4
**Confidence:** 3

**Summary:**

This paper studies decoding schedules for diffusion LLMs (dLLMs) and argues that the commonly used fixed block schedule is semantically agnostic, forcing premature unmasking of low-confidence positions while delaying high-confidence positions just outside the active block, thus yielding a suboptimal quality–speed trade-off. To address this, we propose Dynamic Sliding Block (DSB), a training-free block scheduling method that uses a sliding block with a dynamic size to overcome the rigidity of the naive block. To further improve efficiency, we introduce DSB Cache, a trainingfree KV-cache mechanism tailored to DSB. According to the experimental results presented in the paper, DSB consistently improves both accuracy and decoding speed over fixed-block baselines and standard training-free caching methods on LLaDA and Dream across reasoning and code benchmarks.

**Compliance With Llm Reviewing Policy:**

Affirmed.

**Final Justification:**

Thank you for the detailed rebuttal. The response partially alleviates several of my concerns, and I appreciate the authors’ effort to address the main issues raised in the review.

In particular, the rebuttal provides a clearer explanation of the differences between DSB and AdaBlock, adds comparison results with AdaBlock and dKV-Cache, and includes a more detailed discussion of the relationship to WeDLM. While I still think the distinction from AdaBlock is not yet fully established at the mechanism level, and the newly added experimental results appear to be affected by technical issues that limit how much I can rely on the efficiency numbers, I do think the rebuttal improves the overall clarity and completeness of the work.

Therefore, I am willing to raise my score slightly. That said, I still suggest the authors to strengthen the final version by making the novelty over closely related scheduling methods more explicit and by providing cleaner and more reliable comparisons for the newly added baselines.

**Key Questions For Authors:**

1.The paper currently lacks a sufficiently thorough comparison with WeDLM, a training-based method for the same problem. A more detailed discussion of the relative advantages and limitations, underlying assumptions, and applicability of the two approaches would help clarify the method’s contribution and improve the paper’s credibility.

**Limitations:**

The current limitations of this work are mainly threefold. First, the conceptual distinction from closely related prior work, especially AdaBlock, remains insufficiently clarified, and the manuscript does not clearly establish what is fundamentally novel in DSB beyond the shared motivation. Second, the empirical evaluation is incomplete, as relevant training-free scheduling or decoding baselines, including AdaBlock, are missing. Third, the KV-cache analysis remains limited, since it compares only against Dual Cache while omitting dKV-Cache, a closely related published baseline for diffusion language models.

**Strengths And Weaknesses:**

Strengths:
1.	Technical novelty and innovation: Motivated by the limitations of naive block scheduling, DSB incorporates semantic difficulty into its boundary update rules. The paper also introduces a tailored DSB caching mechanism. By using a refreshed prefix window tied to block movement, together with periodic global refresh, it alleviates the cache instability caused by the sliding schedule.
2.	Experimental rigor and validation: Extensive evaluations on four dLLM models and five benchmarks show overall performance. The paper also includes ablation studies on the initial block length, generation length, maximum block length, and minimum window length.
3.	Clarity of presentation: The problem formulation are clear, and the method is explained in detail with the aid of informative figures and pseudocode.

Weaknesses:

1.Although the paper acknowledges closely related work such as AdaBlock, the conceptual distinction remains insufficiently clarified. In particular, the core motivation behind DSB substantially overlaps with prior observations that fixed block schedules prematurely commit low-confidence positions while delaying high-confidence positions outside the active block. However, the manuscript does not clearly articulate what is fundamentally new in DSB at the mechanism level beyond this shared motivation.

2.Missing key baselines: the paper does not compare against relevant baselines such as AdaBlock, some of which are directly relevant training-free scheduling/decoding methods within the scope of this work.

3.The experimental section claims comparisons along three dimensions, namely decoding strategy, block scheduling, and KV caching. However, the KV caching evaluation is limited to Dual Cache and omits dKV-Cache, which is a closely related and published baseline for diffusion language models. As a result, a key experimental comparison is missing from the caching analysis.

---

> ### Author Rebuttal · Authors · 2026-03-31
>
> Thanks for the insightful and careful comments. All review comments will be taken into consideration in the next revision.
> ### 1. Distinction Between AdaBlock-dLLM and DSB.
> (1). Block scheduling algorithm. AdaBlock identifies valid delimiters (such as "." and ",") and determines the range of the next block by performing an additional forward after the previous block is completely unmasked. However, DSB uses dynamic block scheduling, adjusting the number of blocks after each forward based on the number of decoded blocks in that forward.
>
> (2). KV Cache. AdaBlock's KV cache implementation directly applies the fast-dllm KV cache method. In contrast, DSB Cache is a caching method designed for DSB itself, maintaining high quality by maintaining a prefix window.
>
> ### 2. Comparison with AdaBlock-dLLM.
> The result table is at https://ibb.co/7xGyh0yn. All experiments were conducted under the same setting as the main experiment, the full benchmark evaluation will be added in next revision. From the table, the accuracy appears correct, but the TPS is abnormal. After investigation, it is the CPU of the machine that has a problem. Suspicious results are marked in red. **I will address this issue as soon as possible and report the accurate data during the subsequent rebuttal stage.** Sorry for that, and thank you for your understanding.
>
> However, from an accuracy perspective, DSB still outperforms AdaBlock in most cases, except in the case of a Dream model without a KV cache. In particular, DSB with Cache outperforms AdaBlock by almost 8 in the dream model and humaneval task.
>
> ### 3. The KV Cache Evaluation.
> The result table is at https://ibb.co/B5m9v2VQ. We added dKV-Cache (decode, the most stable mode) as a new baseline and aligned it with the experimental settings, the full benchmark evaluation will be added in next revision. And this evaluation also suffering from the same issues as above.
>
> From an accuracy perspective, DSB is significantly superior to dKV-Cache, only slightly inferior to dKV-Cache in the case of LLaDA with KV Cache on HumanEval. DSB with KV Cache also shows a significant improvement over dKV Cache in Dream model and humaneval tasks, with an improvement of almost 5.
>
> ### 4. Detailed Discussion of WeDLM.
> In terms of timing, WeDLM should be a concurrent project of DSB. The key motivation of WeDLM is the low efficiency of the current block diffusion KV cache and its slower inference speed compared to AR, thus proposing a new training paradigm and an adaptive inference strategy based on the training model. DSB, on the other hand, is based on the observation in AdaBlock that dllm has defects in semantic processing, proposing a new training-free method to improve the inference quality and efficiency of DLM.
>
> (1). Relative advantages and limitations. WeDLM, through its novel training method, improved the performance of dLLM in experimental results. However, its complex training method and high training costs (100B tokens for pretraining, 10B tokens for SFT) make it difficult to use. DSB, as a plug-and-play strategy, also improved the performance of dLLM to some extent, but its drawback is instability: In rare cases, it increases speed but loses minimal accuracy.
>
> (2). Underlying assumption. WeDLM assumes that masked recovery can be effectively achieved under causal attention without requiring full bidirectional interactions and in KV-cached serving, prefix cacheability is more critical than raw parallelism for practical inference speed. DSB assumes that the current predefined fixed-block setting is not optimal, as it ignores the semantic connections among blocks. This assumption has been verified in answer 1 of Reviewer TWPe.
>
> (3). Applicability. With sufficient computing power, WeDLM can train high-performance models, which hold promise for industrial production. DSB is more lightweight and flexible, and can directly and simply improve existing common dLLMs.

---

> > ### Author Rebuttal · Reviewer_C9bf · 2026-04-03
> >
> > Thank you for the detailed rebuttal. The response partially alleviates several of my concerns, and I appreciate the authors’ effort to address the main issues raised in the review.
> >
> > In particular, the rebuttal provides a clearer explanation of the differences between DSB and AdaBlock, adds comparison results with AdaBlock and dKV-Cache, and includes a more detailed discussion of the relationship to WeDLM. While I still think the distinction from AdaBlock is not yet fully established at the mechanism level, and the newly added experimental results appear to be affected by technical issues that limit how much I can rely on the efficiency numbers, I do think the rebuttal improves the overall clarity and completeness of the work.
> >
> > Therefore, I am willing to raise my score slightly. That said, I still suggest the authors to strengthen the final version by making the novelty over closely related scheduling methods more explicit and by providing cleaner and more reliable comparisons for the newly added baselines.

---

> > > ### Author Response · Authors · 2026-04-03
> > >
> > > Thank you for your time and the raised score. I also appreciate your understanding regarding the hardware issue. I have now repeated the experiment on another proper H200 machine. All of your suggestions will be taken into account in the next revision. I hope that the following responses fully address your concerns.
> > >
> > > ### 1. Clearer distinction from AdaBlock-dLLM at the mechanism level.
> > >
> > > Both methods focus on the limitation of predefined, fixed block shceduling, which is **static** and can force premature commitments to uncertain positions while delaying easy positions near block boundaries. However, AdaBlock uses an explicit method to define these static blocks, while DSB uses a **dynamic, real-time adaptive** block.
> > >
> > > (1). Block scheduling difference at mechanism level. As shown in figure at https://ibb.co/h1DWM2LT : (i) AdaBlock determines the next block after each decoding step by detecting delimiters (such as "." and ",") with high confidence.  However, AdaBlock’s blocks are still static in the sense that the next block only begins after the current block has been completely decoded (same with the predefined fixed block). In the figure, based on the forward confidence of step 1, C is estimated to have delimiters such as "\n", so A, B and C form the first block. Only after these are totally decoded will D, E and F may be considered. Thus, at step 2, AdaBlock can only unmask position B, regardless of how confident it is about other positions. (ii). DSB starts from an initial block. After each forward pass, the left and right boundaries are moved based on the result of current forward pass. In the figure, the initial block includes positions A, B, and C, and positions A and C are decoded in the first step. The block then moves and expands: the left boundary shifts two positions to the right (the same number of positions decoded in this step), and the right boundary moves to the first undecoded position. Consequently, DSB can unmask position D rather than B in step 2, allowing it to consider more informative positions earlier.
> > >
> > > AdaBlock may face the following limitations: (i). As illustrated above, AdaBlock’s block scheduling depends on previously predicted delimiters, which may be inaccurate, especially when the model’s confidence is low. With inaccurate delimiter predictions, this method inherits the drawbacks of predefined, fixed blocks: it may still forces premature commitments to uncertain positions while delaying easy positions near block boundaries. In contrast, DSB better adapts to these issues through dynamic, real-time adjustments, largely avoiding the problems inherent to static blocks. (ii). Moreover, this heavy reliance on explicit delimiters may limit the generality of AdaBlock’s approach.
> > >
> > > (2). KV Cache distinction from AdaBlock. DSB proposes a tailored KV cache mechanism (DSB Cache), whereas AdaBlock simply adopts an existing KV cache method without any additional design.
> > >
> > > ### 2. Correction of previous comparisons.
> > >
> > > I have repeated the experiments on another properly configured H200 machine. Thank you again for your previous understanding.
> > >
> > > (1). Correction of comparison with AdaBlock. The result table is at https://ibb.co/ZpjGtG3h. Overall, DSB outperforms AdaBlock in both accuracy and TPS, except for a slight underperformance on the Dream model without a KV cache. Specifically, DSB achieves an improvement of nearly 8 accuracy points on the HumanEval task for the Dream model with KV Cache, and an improvement of almost 22 TPS on the GSM8K task for the LLaDA model with KV Cache.
> > >
> > > (2). Correction of comparison with dKV-Cache. The result table is at https://ibb.co/Wvb033sP. In summary, DSB with DSB Cache significantly outperforms Naive Block with dKV-Cache, except for a slight decrease in accuracy for the LLaDA model on the HumanEval task. Specifically, DSB with DSB Cache improves accuracy by almost 5 points on the HumanEval task for the Dream model, and by nearly 47 TPS on the GSM8K task for the LLaDA model.
> > >
> > > I hope this fully addresses your concerns. Please feel free to ask any further questions.

---

### Decision · Program_Chairs · 2026-04-30

**Decision:**

Accept (regular)

**Comment:**

This paper introduces a method for training diffusion LLMs that replaces the rigid fixed-block schedule. By utilizing a sliding block with dynamic boundaries that adapt based on the confidence of decoded tokens, the method aims to improve the trade-off between generation quality and inference speed. The authors also present a specialized KV-cache mechanism that employs a refreshed prefix window.

Reviewers noted that the work had extensive experiments, was well-written, and addresses an important problem. There were some issues raised in the initial reviews:
- Absence of AdaBlock as an experimental baseline, lacking clarity on distinction, and discrepancies with original paper
- Missing discussion of existing methods incl. dKV-Cache, WeDLM, and WavefrontDiffusion
- Performance gains are inconsistent

Despite the extensive original experiments, the similarity to the mentioned methods does dictate that additional ablations and comparisons should be run. The authors did this and presented results. They added descriptions of how their method differs as well.

Admittedly there are some inconsistent gains with this method, as it does not improve when added to every dLLM architecture. The authors have been upfront about this in the discussion phase, but such limitations must be added to the paper.

I am recommending acceptance. The authors must update their paper with:
- sufficient acknowledgment of related works (dLLMs are a fast developing field, so I give the authors some leeway for not discussing them in the original draft, but now they should be added for the final version)
- complete and reliable experimental results cleaning up what was done during the rebuttal phase (the authors admitted some hardware issues caused inconsistent results during the rebuttal)
- A discussion of the limitations of the method, and honest evaluation of when and where the method does not improve existing results. I believe a method can be useful to the community even if results are not uniformly positive, as long as the evaluations are complete and transparent about such limitations.